# Potential for Therapeutic-Loaded Exosomes to Ameliorate the Pathogenic Effects of α-Synuclein in Parkinson’s Disease

**DOI:** 10.3390/biomedicines11041187

**Published:** 2023-04-17

**Authors:** David J. Rademacher

**Affiliations:** Department of Microbiology and Immunology and Core Imaging Facility, Stritch School of Medicine, Loyola University Chicago, Maywood, IL 60153, USA; drademacher@luc.edu; Tel.: +1-708-216-3395

**Keywords:** exosomes, extracellular vesicles, Parkinson’s disease, pathogenesis, therapeutics, α-synuclein, neurodegeneration, genetic modification, chemical modification

## Abstract

Pathogenic forms of α-synuclein (α-syn) are transferred to and from neurons, astrocytes, and microglia, which spread α-syn pathology in the olfactory bulb and the gut and then throughout the Parkinson’s disease (PD) brain and exacerbate neurodegenerative processes. Here, we review attempts to minimize or ameliorate the pathogenic effects of α-syn or deliver therapeutic cargo into the brain. Exosomes (EXs) have several important advantages as carriers of therapeutic agents including an ability to readily cross the blood–brain barrier, the potential for targeted delivery of therapeutic agents, and immune resistance. Diverse cargo can be loaded via various methods, which are reviewed herein, into EXs and delivered into the brain. Genetic modification of EX-producing cells or EXs and chemical modification of EX have emerged as powerful approaches for the targeted delivery of therapeutic agents to treat PD. Thus, EXs hold great promise for the development of next-generation therapeutics for the treatment of PD.

## 1. Introduction

Extracellular vesicles (EVs) are membrane-enclosed particles released by cells into the extracellular space [1,2]. EVs can be classified as exosomes (EXs), microvesicles, or apoptotic bodies based on their origin and size [1,2,3]. EXs are enclosed within a single phospholipid bilayer, secreted by all cell types, formed by the inward invagination of the endosomal membrane and fusion of the multivesicular body (MVB), and are typically 30–150 nm in diameter [1,3,4,5]. Microvesicles are EVs that form from direct outward budding from the cell’s plasma membrane and are typically 100 nm to 1 µm in diameter [1,2,3,4,5]. Although the route of microvesicle formation is not fully understood, it is thought to require cytoskeleton components, molecular motors, and fusion machinery [6]. Apoptotic bodies are EVs formed during apoptosis and released into the extracellular space; they range in diameter from 50 nm to 5 µm [3]. Apoptotic bodies form through the separation of the cell’s plasma membrane from the cytoskeleton due to increased hydrostatic pressure after the cell contracts [7]. EVs contain thousands of different biologically active molecules, including nucleic acids, proteins, lipids, and metabolites [8,9,10,11]. Here, we focus on the most extensively studied EVs, typically designated as EXs, which play key roles in intercellular communication by delivering biologically active cargo to recipient cells, thereby altering the recipient cell’s functions [12,13]. Thus, EXs hold great promise for developing next-generation delivery vehicles of therapeutic agents.

## 2. Role of EXs in the Pathogenesis of Parkinson’s Disease

Parkinson’s disease (PD) is the second most common neurodegenerative disease in the world after Alzheimer’s disease, affecting 1–2% of the population over the age of 65 [14]. There are approximately seven million PD cases in the world; approximately one million of those cases are in the United States [15]. As the population ages, the burden on society attributable to PD is expected to increase substantially. The main pathological changes in PD are a progressive loss of dopamine (DA)-secreting neurons in the substantia nigra, a significant decrease of DA in the striatum, and the appearance of eosinophilic inclusions in the cytoplasm of DA neurons in the substantia nigra, namely Lewy bodies (LBs) [16]. The progressive loss of nigrostriatal neurons leads to the appearance of classical parkinsonian motor symptoms (e.g., bradykinesia, tremor, and rigidity) and numerous non-motor symptoms (e.g., depression, constipation, pain, gastrointestinal dysfunction, and sleep problems) [17,18]. The presence of α-synuclein (α-syn) aggregates in LBs [16], approximately 90% of which are phosphorylated on serine residue 129 [19], and the finding that mutations in the α-syn gene, *SNCA*, cause familial PD [20,21,22,23] and accelerate the pathogenic aggregation of α-syn [24,25], strongly suggested a role for α-syn in the pathogenesis of PD.

Prions are infectious agents in which the conformationally altered protein, PrP^Sc^, recruits and corrupts its counterpart protein, PrP^C^, generating self-propagating, misfolded species that spread from cell to cell [26]. According to the prion hypothesis of PD [27], like prion proteins, misfolded α-syn is transmitted from diseased cells to healthy cells, thereby spreading α-syn pathology in the PD brain [28,29]. The notion that EXs can be used as a carrier of toxic, misfolded proteins, such as α-syn, is an important tenet of the prion hypothesis of PD and is well supported by evidence. In vitro experiments provided the first evidence that newly synthesized monomeric and aggregated α-syn was released into the extracellular environment [30,31,32], a finding consistent with the presence of α-syn in human cerebrospinal fluid and blood plasma in both PD and normal human subjects [33,34]. Interestingly, EXs provide an environment conducive to α-syn aggregation [35]. In vitro studies have demonstrated α-syn release in EXs from donor neurons, uptake by recipient neurons, and subsequent cell death of recipient neurons [36,37,38]. When EXs harvested from the brain tissue of dementia with Lewy bodies patients were injected into the brains of mice, α-syn was taken up by neurons and astrocytes, and intracellular α-syn accumulation was observed [39]. Additional support for the prion hypothesis of PD comes from a study that examined the EXs isolated from the serum of PD patients, which contained a higher content of α-syn phosphorylated at serine residue 129 and oligomeric and monomeric α-syn than controls. In vitro studies demonstrated that the PD EXs, which contained an abundant amount of toxic, misfolded α-syn, were taken up by recipient cells, and acted as a seed or template to induce the aggregation of endogenous α-syn in recipient neurons. Interestingly, in human midbrain DA neuron cell cultures, pathogenic, misfolded α-syn was secreted in EXs via an autophagic secretory pathway [40]. Moreover, PD EX administration to mice resulted in DA neuron degeneration, microglial cell activation, and motor deficits [41]. Notably, neuron-to-neuron, neuron-to-microglia, microglia-to-neuron, neuron-to-astrocyte, and astrocyte-to-neuron transfer of α-syn has been demonstrated (Figure 1) [42,43,44,45,46].

## 3. Role of Microglia in the Pathogenesis of PD

Microglia, the main resident immune cells in the brain, can have beneficial and harmful effects on PD depending, in part, on their activation state. Concerning their harmful effects on PD, microglia have been implicated in the pathogenesis of PD. Positron emission tomography imaging demonstrated microglia activation in the substantia nigra and striatum of PD patients [47]. Increased microglia activation in the midbrain was correlated with a loss of DA-secreting nerve terminals in the striatum [48]. In addition, activated microglia were more frequently observed near LBs containing α-syn and near dying neurons [49].

Concerning their beneficial effects, microglia can enhance neuronal survival by releasing trophic factors, clearing debris, dead cells, and misfolded α-syn aggregates in PD [50,51,52,53]. However, α-syn activates microglia (Figure 1) [53,54]. Activation of microglia induces an oxidative stress response, including the release of reactive oxygen species (ROS) and nitric oxide, the production of NADPH oxidase, and the release of pro-inflammatory cytokines and chemokines (Figure 1) [55,56,57]. Inflammation and oxidative stress can lead to neuron dysfunction and cell death (Figure 1) [58,59], effects that have been linked to the pathogenesis of PD [60], and it is known that microglia release EXs [61]. There is evidence to support the notion that α-syn can be transferred from microglia to neurons via EXs and induce α-syn aggregation in the recipient neurons (Figure 1), an effect that is exacerbated by microglia-derived pro-inflammatory cytokines. In addition, microglial EXs isolated from PD patients induced α-syn aggregation and cell-to-cell transfer of α-syn, DA neuron degeneration in the substantia nigra, and motor deficits in mice [45].

## 4. Role of Astrocytes in the Pathogenesis of PD

In common with microglia, astrocytes have beneficial and harmful effects on PD. Astrocytes can enhance neuronal survival by releasing trophic factors, antioxidants, and other factors that protect against oxidative stress [62,63,64]. Although a consensus has not been reached on the degree of astrocyte activation in the PD brain [65,66], α-syn aggregates have been observed in human astrocytes [67], and there is considerable evidence suggesting that α-syn activates astrocytes that, in turn, result in the astrocytic release of pro-inflammatory cytokines, chemokines, and ROS, microglial cell activation, and neuronal cell death (Figure 1) [53,68,69].

## 5. Pathogenic α-Syn-Containing EXs as Therapeutic Targets

Given the abundance of evidence implicating α-syn-containing EXs in the pathogenesis of PD, a logical therapeutic approach for PD is to minimize or eliminate the pathogenic effects of α-syn-containing EXs. This could be accomplished by decreasing EX biogenesis in parent cells, removing pathogenic EXs from circulation, and inhibiting EX uptake by the recipient cells (Figure 2A–D).

### 5.1. Decreasing EX Biogenesis

Several proteins responsible for EX biogenesis have been identified as targets to decrease pathogenic EX formation. EXs are formed by the invagination of the MVB system and their fusion with the plasma membrane [71]. As EX formation requires either endosomal sorting complexes required for transport (EXCRT)-dependent or ESCRT-independent cargo sorting at the MVB and MVB-plasma membrane fusion, related proteins can be regarded as potential therapeutic targets (Figure 2A) [72]. Two extensively studied proteins are the ALG-2-interacting protein X (ALIX) and the Rab protein [73,74,75]. During EX biogenesis, ALIX proteins are associated with the invagination of the MVB membrane by recruiting ESCRT proteins. Treatment with ALIX small interfering RNA (siRNA) and siRNA directed against the ALIX ligand, syntenin, suppressed ALIX function, resulting in reduced EX biogenesis [73]. Rab27a and Rab27b are notable as they are involved in the process of MVB fusion with the plasma membrane [71,76]. Knockdown or silencing of Rab27a and Rab27b reduced the number of EXs released [77]. In addition, the inhibition of two Rab27 effectors, Slp4 and Slac2b, also reduced the number of EXs released [76]. GW4869 is a potent neutral sphingomyelinase inhibitor that blocks EX production by preventing the formation of intraluminal vesicles (ILVs) (Figure 2A) [78]. Pretreatment of α-syn-activated microglia with GW4869 decreased the release of cathepsin L-containing EXs from microglia, which prevented neuronal death [79]. Similarly, treatment with GW4869 decreased EX release by activated microglia and prevented the death of DA neurons in midbrain slice cultures [80]. Systemic administration of DDL-112, an inhibitor of neutral sphingomyelinase, decreased EX biogenesis, reduced the number of α-syn aggregates in the substantia nigra, and improved motor function in an α-syn mouse model of PD [81].

### 5.2. Depleting Circulating Pathogenic EXs

After EXs are released from parent cells, they are either taken up by neighboring cells or travel to distant recipient cells to deliver their cargo. One interesting strategy to deplete pathogenic EXs from circulation is to use EX-specific antibodies so that EXs can be removed by the immune system (Figure 2B). The administration of anti-CD9 and anti-CD63 antibodies resulted in phagocytosis of the antibody-bound EXs by macrophages (Figure 2B) [82].

### 5.3. Inhibiting EX Uptake by Recipient Cells

In an attempt to ameliorate EX-mediated pathogenic cell-to-cell communication, researchers have inhibited EX uptake by recipient cells (Figure 2C) [83,84]. Endocytosis inhibitors have been heavily studied as potential therapeutics, as EXs are primarily taken up by recipient cells via endocytosis [70]. Cytochalasin D inhibits phagocytosis and endocytosis by blocking actin polymerization and inducing depolymerization of actin filaments [83]. Cancer-associated fibroblast-derived EXs were not effectively taken up by cancer cells in the presence of cytochalasin D [83]. Dynasore blocked the uptake of cancer cell-derived EXs due to an endocytosis-inhibiting effect [84]. In addition, the destabilization of lipid rafts in the plasma membrane is another strategy for inhibiting EX uptake (Figure 2C) [85,86].

## 6. The Therapeutic Effects of Stem Cells Are Mediated by EXs

In recent years, some non-pharmacological methods, such as gene therapy and stem cell therapy, have been considered potential therapeutics for neurodegenerative diseases including PD [87,88]. Mesenchymal stem cells (MSCs) are multipotent progenitor cells that can be isolated from a wide variety of tissues (e.g., bone marrow, adipose tissue, dental tissues, skin, salivary gland, and limb buds) [89]. MSCs are considered therapeutic agents due to their effects on several biological processes, such as immune regulation, oxidative stress, and cytokine secretion [90]. For example, MSCs exert significant antioxidant effects in neurodegenerative diseases [91,92]. While MSC transplantation has been employed in the treatment of several diseases, such as cancer, nerve injury, and neurodegeneration [93,94,95], several studies have shown that MSC transplantation may cause tumors, embolisms, and abnormal cell differentiation [96], limiting the clinical translation of MSC transplantation as a therapy for PD. Importantly, MSCs exert their biological effects mainly by the secretion of EXs. Thus, the use of EXs derived from MSCs retains the therapeutic potency of MSCs, while preventing the possible damage caused by MSCs [97].

Bone marrow-derived stem cells (BMSCs), in common with other stem cells, can be differentiated into different cells under different physiological conditions. BMSCs can selectively migrate to a site of damage, and interact with neurons and glia, where they stimulate the production of growth factors, such as brain-derived neurotrophic factor (BDNF) and nerve growth factor [98,99]. BMSCs have beneficial effects in models of neurodegenerative diseases [100]. For example, the injection of BMSC-derived EXs into the DA-depleted striatum improved parkinsonian behavior, tyrosine hydroxylase expression, and decreased protein levels of interleukin-6, interleukin-1β, tumor necrosis factor-α, and ROS in the substantia nigra in a rat PD model [101]. It is known that BMSCs mediate their effects through paracrine activities [102]. Importantly, the paracrine activities of BMSCs are mediated through EXs [103].

## 7. EXs as Therapeutic Delivery Systems in PD

The first-line treatment for PD is the administration of DA and/or by administering agents that increase DA in the brain, specifically, the striatum. Although DA-replacement therapy benefits many PD patients, its therapeutic window is limited due to its decreasing efficacy and increasing side effects, such as dyskinesias [104,105]. Importantly, delivering DA to the brain or agents that increase DA in the brain is difficult due to the blood–brain barrier (BBB). For example, although L-3,4-dihydroxyphenylalanine (L-DOPA) is the most effective treatment for PD symptoms, approximately 1% of the L-DOPA administered systemically reaches the brain [106]. After L-DOPA has reached the brain, it must be converted to DA by DOPA decarboxylase, which is less active in the brains of patients with PD [107]. Moreover, long-term administration of L-DOPA is marred by the emergence of abnormal involuntary movements called L-DOPA-induced dyskinesias [106].

EXs have the potential to serve as carriers of therapeutic agents into the diseased PD brain, in part, due to their ability to readily cross the BBB [108,109], the potential for targeted delivery of exosomal cargo over long distances, and immune resistance [110]. The intravenous administration of DA-encapsulated blood EXs readily crossed the BBB and delivered DA to the brain, including the striatum and substantia nigra. DA-encapsulated EXs increased brain DA content by greater than fifteen-fold and resulted in motor behavioral improvements and increases in DA synthetic enzymes and enzymes against oxidative stress in a 6-hydroxydopamine (6-OHDA) model of PD. Importantly, compared to the intravenous administration of free DA, DA-encapsulated EXs had greater therapeutic efficacy and lower toxicity [111]. Intranasal administration of catalase-loaded EXs was neuroprotective in a 6-OHDA model of PD [112]. The administration of MSC-derived EXs rescued DA neurons in a 6-OHDA model of PD [113]. Stem cell-derived EXs carry beneficial microRNAs (miRNAs) that reduce neuroinflammation in animal models of PD. For example, miR-133b, one of the miRNAs downregulated in PD, can promote neurite outgrowth in both in vitro and in vivo models of PD [114]. In addition, EXs isolated from human neural stem cells (NSCs) exerted a protective effect on PD pathology in a 6-OHDA in vitro and an in vivo mouse model of PD by reducing intracellular ROS and counteracting the activation of apoptotic pathways. NSC-derived EXs carry anti-inflammatory factors and specific miRNAs (i.e., has-miR-182-5p, has-miR-183-5p, has-miR-9, and has-let-7) involved in cell differentiation that contributed to decreased cell loss [115].

## 8. Strategies to Load EXs with Therapeutic Cargo

After EXs are isolated from tissues, body fluids, or cell culture medium by differential or gradient ultracentrifugation, co-precipitation, size exclusion chromatography, or field flow fractionation [116] then purified to remove unwanted material, they can be loaded with cargo. Notably, EXs are endowed with an aqueous core and a lipid bilayer that allow both hydrophilic and lipophilic cargo to be loaded [117]. In addition to the delivery of small therapeutic compounds, EXs have a natural capacity to transport siRNA, short hairpin RNA (shRNA), miRNA, and proteins [118]. Strategies for loading cargo into EXs include incubation, transfection, and physical treatments.

### 8.1. Incubation

#### 8.1.1. Incubation of Desired Cargo with EXs

The simplest way to load cargo into EXs is to incubate the desired cargo with EXs or EX-secreting cells to allow the cargo to diffuse into the EXs, following its concentration gradient. Several types of cargo, such as small molecule drugs, nucleic acids, proteins, and peptides have been loaded into EXs using this method [119,120,121]. Notably, BDNF has been loaded into macrophage-derived EXs and delivered into the brain [120] while the anti-inflammatory and anti-oxidative stress agent, co-enzyme Q10, has been loaded into EXs obtained from adipose-derived stem cells [121]. The strength of the incubation strategy is that it is technically easy and has minimal effects on the structural integrity of the EXs. However, the loading efficiency is low, and the amount of cargo loaded is difficult to control due to the physical and chemical properties of the cargo and EX. For example, hydrophilic drugs tend to reside in the aqueous phase of the interior of EXs, while hydrophobic drugs are more stable in the EX lipid bilayer [122,123]. In addition, pH can influence loading efficiency. When the hydrophilic compound, doxorubicin, was loaded into macrophage-derived EXs, a pH of 8.0 facilitated the diffusion of the compound across the EX lipid bilayer [124].

#### 8.1.2. Incubation of Desired Cargo with EX-Secreting Cells

Drugs and nanomaterials were incubated with EX-secreting cells to generate cargo-loaded EXs. Some small molecule drugs directly pass across the lipid bilayer of parent cells, are packaged into ILVs, and then secreted as EXs. For example, macrophages were incubated with curcumin to generate curcumin-loaded EXs, which were able to cross the BBB and enter the brain [125]. In addition, nanomaterials were incubated with EX-secreting cells to generate cargo-loaded EXs. Although they may induce autophagy and may be destroyed in lysosomes, undegraded nanomaterials are exocytosed within EXs [126]. For example, doxorubicin-loaded silicone nanoparticles were incubated with cancer cells to obtain nanoparticle-loaded EXs for the treatment of lung cancer [126].

### 8.2. Transfection or Transduction

Transfection or transduction is the most common strategy for stably loading nucleic acids, proteins, and peptides into EXs. Using transfection reagents, specific plasmids are transduced into cells to ectopically express the desired nucleic acids, proteins, or peptides that are later packaged into EXs. For example, MSCs have been transfected with a miR-122-expressing plasmid using a Lipofectamine-based protocol to generate miR-122-enriched EXs [127]. HEK293 cells have been transduced with designed plasmids to generate catalase mRNA-loaded EXs that target the brain to treat PD [128]. HEK293 cells were transfected with lentivirus to generate EXs loaded with translocase of the outer mitochondrial membrane 40 (Tom40). EX-mediated delivery of Tom40 protected cells against hydrogen peroxide-induced oxidative stress [129]. Other types of cells can be transfected with vectors that express proteins and peptides to generate protein- or peptide-loaded EXs [130,131]. In addition, EXs can be directly transfected with nucleic acids by chemical treatment. HEK293 cells have been transfected with siRNA by a heat-shock protocol [132] and cell-derived EXs have been transfected with miR-497 and miR-126 by commercially available kits [133,134]. Although transfection is a common strategy for loading nucleic acids, proteins, or peptides into EXs, the loading efficiency is low [135] and direct chemical transfection of EXs introduces impurities [136].

### 8.3. Physical Treatments

Physical treatments produce micropores in the EX membrane or membrane recombination that promotes the entry of cargo into EXs to achieve cargo-loaded EXs. Physical treatments include sonication, electroporation, extrusion, the freeze-thaw method, incubation with membrane permeabilizers, and dialysis.

#### 8.3.1. Sonication

Sonication is a physical strategy that applies an extra mechanical shear force to weaken the EX membrane, which promotes the loading of EX cargo [137]. Cancer cell-derived EXs incubated with the anti-cancer drug, gemcitabine, were sonicated. The loading capacity of gemcitabine-loaded EXs was more than four times greater than that of those obtained using the incubation approach [137]. Other researchers have reported that the sonication method results in a higher loading capacity than the incubation approach (e.g., [124]). In addition, nanoparticles and catalase have been loaded into EXs via sonication [112,138]. Note, however, that the sonication method has the potential to produce significant membrane damage. For example, a significant decrease in EX membrane microviscosity was observed after sonication, an effect that was completely reversed after incubating the EXs for 1 h at 37 °C after sonication [139]. Thus, sonication is a simple and effective method for loading cargo into EXs with high loading capacity.

#### 8.3.2. Electroporation

Electroporation is a strategy for loading cargo into EXs through the use of an extra electrical field that produces micropores on the EX membrane to increase permeability. Drugs, nucleic acids, and nanomaterials have all been loaded into EXs using electroporation [139,140,141]. Although drugs can diffuse, in accordance with their concentration gradient, into EXs via the incubation method, the use of electroporation can significantly increase drug loading efficiency. Researchers have developed modified dendritic cell-derived EXs, which specifically target the brain, by introducing a brain targeting peptide, rabies virus glycoprotein (RVG), on the exterior surface of the EX. An anti-α-syn short hairpin RNA-minicircle (shRNA-MC) construct was loaded into RVG EXs via electroporation. Intravenous administration of shRNA-MC-loaded EXs decreased α-syn aggregation, attenuated the loss of DA-secreting neurons, and improved clinical symptoms in an α-syn preformed fibril model of PD [142].

#### 8.3.3. Extrusion

Extrusion is a physical procedure that utilizes a syringe-based extruder and mechanical force. In this approach, the cargo and EXs are loaded into the extruder equipped with a porous membrane. The extrusion process causes the EXs membrane to collapse and blend with cargo to form cargo-loaded EXs after repeated extrusions under specific parameters [143]. This approach has been taken to load the antioxidant, catalase, into EXs and then deliver the catalase-loaded EXs to the brain as a potential anti-PD therapeutic. The extrusion method resulted in high loading efficiency, sustained release of catalase, and protection of the catalase cargo from degradation by proteases. When administered to mice intranasally, a considerable amount of catalase was detected in the brain and the catalase-loaded EXs had a neuroprotective effect in a 6-OHDA model of PD [112].

#### 8.3.4. Freeze-Thaw Method

The first step of this method is to incubate the isolated EXs with the to-be-loaded cargo for a specific amount of time at room temperature. Next, the EX and cargo solution is rapidly frozen at −80 °C or below, then the solution is thawed at room temperature [112]. For better cargo loading, the aforementioned process is repeated for at least three cycles. Although the freeze-thaw approach is simple and effective to load various cargo (e.g., drugs, proteins, and peptides) into EXs, it has a lower cargo-loading capacity than the sonication and extrusion methods [144] and multiple freeze-thaw cycles could inactivate proteins and induce EX aggregation.

#### 8.3.5. Incubation with Membrane Permeabilizers

Saponin is a surfactant molecule that can form complexes with cholesterol in cell membranes and generate pores, thus leading to an increase in membrane permeabilization [145]. Membrane permeabilizers significantly increase the loading capacity of a variety of cargo into EXs, at least compared to the incubation method [144]. Incubation with saponin resulted in an eleven-fold greater loading of a hydrophilic compound compared to the incubation method [122]. Given the concerns about the hemolytic activity of saponin [145], the concentration of saponin during drug loading should be low, and the EXs should be purified after incubation with saponin.

#### 8.3.6. Dialysis

This method involves placing a mixture of cargo and EXs onto dialysis membranes, which are dialyzed by stirring to obtain cargo-loaded EXs. Compared to the incubation approach, the dialysis procedure has increased the amount of cargo loaded into EXs more than eleven-fold [122]. In addition, the dialysis system can be used to reduce the intra-exosomal pH gradient to generate a pH gradient between the inside and the outside of the exosomal membrane [146]. Although the pH gradient modification increases the loading of miRNA and siRNA into EXs, it may induce the degradation of proteins and peptides [146]. Whereas some studies have reported good cellular uptake of cargo-loaded EXs obtained via the dialysis method [147], others have reported poor cellular uptake [122]. Although the dialysis method appears to be a relatively simple and effective EX-cargo loading technique, researchers should carefully consider the type of loading cargo and whether to use a pH gradient modification.

#### 8.3.7. Comparing Different Loading Methods

The advantages and disadvantages of the different loading approaches are given in Table 1. Note that loading compounds in EXs, regardless of the loading method, may result in greater stability, increased bioavailability, and reduced immunogenicity, as well as preserving the activity of the cargo (as the cargo is protected from degradation) [112,148]. The packaging of the hydrophobic compound, curcumin, in EXs substantially increased its stability in aqueous solutions. The solubility of curcumin-loaded EX was five-fold higher than free curcumin.

Although studies directly comparing exosomal loading by the different loading methods are sparse, the exosomal loading efficiency and cellular uptake of catalase across different loading methods have been assessed. The loading efficiency of catalase into EXs by incubation, the freeze-thaw method, incubation in the presence of saponin, sonication, and extrusion were 4.9%, 14.7%, 18.5%, 22.2%, and 26.1%, respectively. The uptake of catalase-loaded EXs by PC12 cells by incubation, the freeze-thaw method, and sonication were 10%, 15%, and 40%, respectively [112]. Interestingly, regardless of the loading method, the cellular uptake of catalase-loaded EXs was substantially greater than that of poly(lactic-co-glycolic acid) nanoparticles [112], which have been used for the delivery of L-DOPA to the brain to treat PD [149]. The loading efficiencies of the small molecule, porphyrin, into EXs by incubation, electroporation, extrusion, incubation in the presence of saponin, and dialysis were compared. Compared to the loading efficiency of porphyrin into EXs by incubation, the loading efficiency of porphyrin into EXs was increased more than eleven-fold by incubation in the presence of saponin and dialysis but not by electroporation or extrusion [122]. These researchers also observed a four-fold increase in drug uptake by breast cancer cells for EXs loaded by incubation in the presence of saponin and electroporation compared to the uptake of drugs not loaded into EXs [122]. When electroporation and sonication were used to load the highly hydrophobic compound, paclitaxel, into EXs, more than 3 times and more than 19 times, respectively, of paclitaxel was loaded into EXs than when the incubation method was used [139].

## 9. Strategies to Target EXs to the Brain

### 9.1. Exploit EX Homing/Tropism

Cells of different origins are known to home in on specific locations in vivo. For example, immune cells preferentially target sites with immunological activity, such as the spleen, to a greater extent than control cells [150]. There is support for the idea that EXs possess intrinsic tropisms based on their cells of origin [151], an attribute that decreases the probability of off-target effects and can be exploited for organ-targeted delivery of EX cargo. For example, EXs secreted from cortical neurons preferentially bind and are endocytosed by neurons [152]. In addition, systemic administration of NSC-derived EXs resulted in preferential brain targeting whereas systemic administration of MSC-derived EXs did not [153]. Notably, brain endothelial cell-derived EXs crossed the BBB and delivered anti-cancer drugs to brain tumors [154].

The currently available methods for EX engineering can be classified into two main approaches: (1) genetic engineering, and (2) chemical modification. Genetic engineering is effective for displaying genetically engineered proteins on the surface of EXs, although it is limited to genetically encodable peptides and proteins. The chemical modification approach can be used to functionalize EXs with a wide range of molecules by using noncovalent or covalent interactions. However, this approach is challenging because of the complexity of the EX membrane and the issues associated with separating unreacted chemicals from the EXs [155].

### 9.2. Genetic Engineering

One interesting approach to target EXs to the brain has been to genetically modify EX-producing cells by transfecting genes expressing a targeting moiety (e.g., peptides, receptors) with exosomal membrane components, such as tetraspanins, lysosomal membrane-associated protein 2B (LAMP2B), or the C1C2 domain of lactadherin [156,157]. The cells transfected with these vectors generate surface-modified EXs that express the targeting moieties via the natural EX biogenesis process. The EXs produced from genetically engineered cells stably display the introduced target moiety on their surface [158]. For example, cells were transfected with a fusion protein comprised of LAMP2B and RVG, and the cells generated EXs with RVG embedded in the exosomal membrane. These RVG-expressing EXs more readily localized to the brain due to the cell surface expression of receptors for RVG by neurons and glia [159]. Intravenous administration of RVG-expressing EXs resulted in a two-fold greater accumulation of EXs in the brain and a substantial accumulation of EXs in the heart and muscle, which also express receptors for RVG (i.e., nicotinic acetylcholine receptors) [151]. Notably, intravenous administration of a slightly modified RVG peptide, RVG-9R, has been used to transport siRNA to neurons to produce a neuron-specific knockdown [160]. In addition, intravenous administration of RVG-expressing EXs loaded with *GAPDH* siRNA specifically delivered the siRNA to neurons and glial cells in the brain, resulting in an approximately two-fold knockdown of GAPDH mRNA compared to non-treated mice [141]. In an attempt to reduce the expression of mutant huntingtin (mHTT) protein, the root cause of Huntington’s disease, mice received tail vein injections of a plasmid containing an RVG, LAMP2B, and mHTT siRNA under the control of a cytomegalovirus promoter. When the plasmid was taken up by hepatocytes, the cytomegalovirus promoter directed the localization of the RVG tag to the EX surface. The RVG-tagged, mHTT siRNA penetrated the BBB, was delivered to the cerebral cortex and striatum, decreased levels of mHTT protein and toxic aggregates in the cerebral cortex and striatum, and ameliorated behavioral deficits in three mouse modes of Huntington’s disease [161].

### 9.3. Chemical Modification

The surface of EXs can be directly engineered via chemical modifications for inducing targetability of therapeutic EXs. One approach is to use covalent attachments of targeting moieties, such as click chemistry, and the other approach uses non-covalent modifications [162].

#### 9.3.1. Covalent Modification of the Surface of EXs

Click chemistry utilizes covalent interactions between an alkyne and azide residue to form a stable triazole linkage, which can be applied to attach targeting moieties on the surface of EXs [162,163]. One of the most common examples of a chemical conjugation method that uses covalent attachments is the modification of the EX’s surface with branched polyethylene glycol (PEG), termed PEGylation [164]. Interestingly, to target sigma receptor overexpressing lung cancer cells, EXs were modified with an aminoethylanisamide-PEG moiety, which served as a targeting ligand for the sigma receptor [165]. In addition, c(RGDyK), a peptide that has a high affinity for integrin α_v_β_3_, which is expressed in reactive cerebral vascular endothelial cells after ischemia, was conjugated to the surface of MSC-derived EXs via click chemistry [166]. c(RGDyK)-labeled EXs exhibited an eleven-fold tropism to the lesioned region of the ischemic brain compared to scrambled c(RGDyK) peptide-labeled EXs [166]. We are not aware of the use of a covalent modification of the surface of EXs for the targeted delivery of therapeutic EXs to the brain to treat PD.

#### 9.3.2. Non-Covalent Modification of the Surface of EXs

The exosomal membrane can also be engineered via non-covalent methods, such as receptor–ligand binding, electrostatic interaction, and hydrophobic insertion [167,168]. Transferrin was used to conjugate superparamagnetic magnetite colloidal nanocrystal clusters to the surface of EXs by binding to transferrin receptors expressed on the EXs [169]. The electrostatic interaction approach to conjugate targeting moieties to EXs involves interactions of cationic species with negatively charged functional groups on the EX membrane [168]. This method has been used to attach cationic lipids and a pH-sensitive fusogenic peptide to the negatively charged membrane of EXs [170]. These fusogenic peptide-expressing EXs exhibited increased binding to the endosomal membrane after endocytosis, which facilitated the intracellular delivery of cargo [170]. The substance 1,2-dioleoyl-sn-glycero-3-phosphoethanolamine-N-hydroxysuccinimide (DOPE-NHS) is a hydrophobic chemical that can be used to conjugate targeting peptides into exosomal membranes. For targeting EXs to the heart, stem cell-derived EXs were conjugated with cardiac homing peptide via a DOPE-NHS linker, which resulted in EX accumulation in the heart [171]. We are not aware of the use of receptor-ligand binding, electrostatic interaction, and hydrophobic insertion methods for the targeted delivery of therapeutic EXs to the brain to treat PD.

## 10. Conclusions

EXs play key roles in intercellular communication by delivering biologically active cargo to nearby or distant recipient cells. The cargo delivered by EXs can have a harmful or beneficial effect on the recipient cell. EXs spread α-syn pathology in the olfactory bulb and the gut, then throughout the PD brain, by transferring pathogenic, misfolded forms of α-syn from diseased cells to healthy cells. Pathogenic, misfolded forms of α-syn are transferred via EXs to and from neurons, astrocytes, and microglia. This sets in motion a cascade of events whereby astrocytes and microglia are activated and then secrete ROS and pro-inflammatory cytokines and chemokines into the extracellular space, which contributes to the degeneration of neurons (Figure 1). Researchers have attempted to minimize or ameliorate the pathogenic effects of α-syn-containing EXs by (a) targeting proteins that play a role in EX biogenesis; (b) developing methods aimed at the removal of α-syn-containing EXs from circulation; (c) inhibiting EX uptake by recipient cells; and (d) loading EXs with therapeutic cargo and delivering them to the brain (Figure 2). Advantages of EXs as carriers of therapeutic agents into the diseased brain include their ability to readily cross the BBB, their potential for targeted delivery of therapeutic cargo over a long distance, and their immune resistance. In addition, a wide variety of cargo, including hydrophilic and lipophilic small therapeutic compounds, siRNA, miRNA, and proteins can be loaded into EXs. The choice of loading method depends on the objective of the study and an assessment of the advantages and disadvantages of each (Table 1). Regardless of the loading method, the loading of therapeutic agents into EXs often results in greater stability, increased bioavailability, protection (e.g., from degradation), and reduced immunogenicity. Genetic modification of EX-producing cells and/or EXs and chemical modification of EXs have emerged as powerful approaches for the targeted delivery of therapeutics to neurons and/or glia. Thus, EXs hold great promise for the development of next-generation therapeutics for the treatment of PD.

## Figures and Tables

**Figure 1 biomedicines-11-01187-f001:**
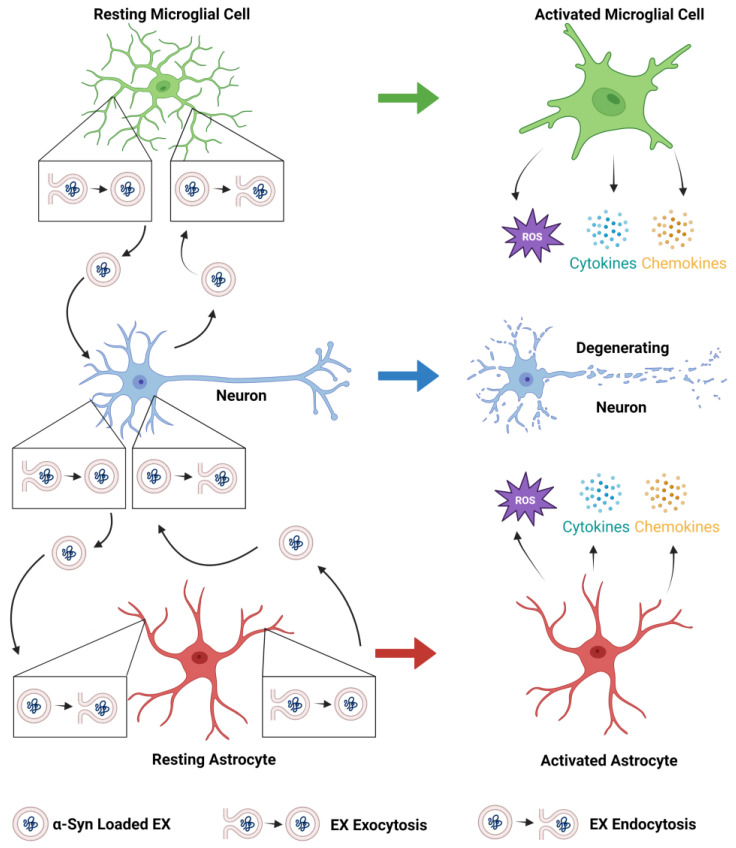
The transfer of α-syn to and from neurons, astrocytes, and microglia. The transfer of α-syn to astrocytes and microglia results in their activation. Activated astrocytes and microglia release ROS, pro-inflammatory cytokines and chemokines, which contribute to the neurodegenerative processes in PD. The figure was created with BioRender.com https://app.biorender.com/ (accessed on 24 March 2023).

**Figure 2 biomedicines-11-01187-f002:**
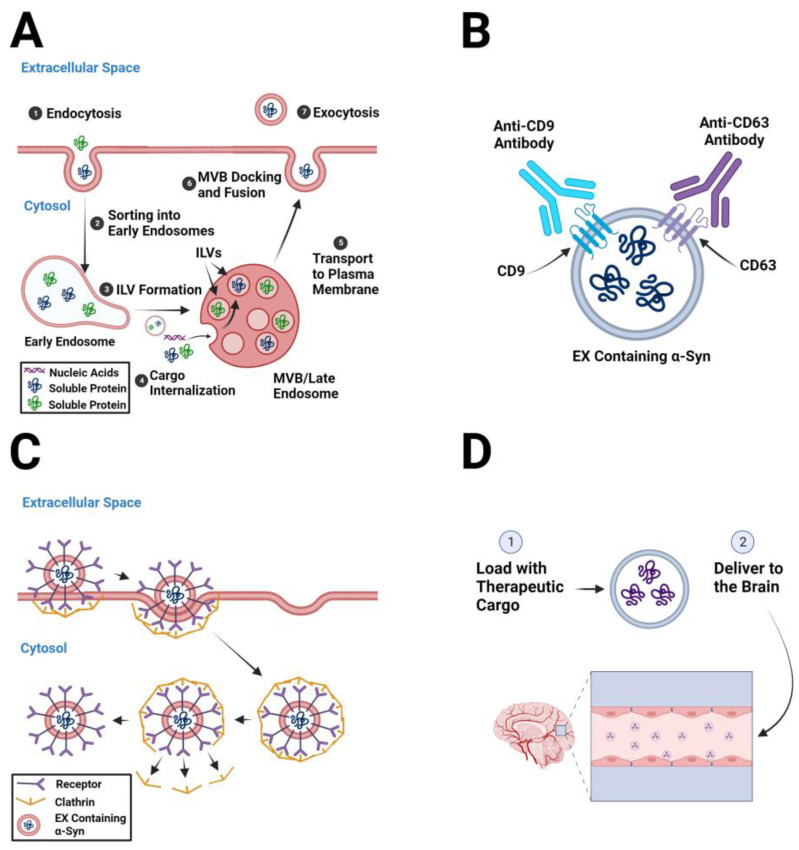
Approaches to minimize or eliminate the pathogenic effects of α-syn-containing EXs in PD. (**A**). The major steps in the biogenesis of α-syn-containing EXs. Therapeutic approaches may target key proteins involved in each of these steps. (**B**). An EX that contains pathogenic, misfolded α-syn and expresses the tetraspanins, CD9 and CD63, is sequestered by antibodies directed against CD9 and CD63 and then cleared from circulation. (**C**). α-Syn-containing EXs are taken up by recipient cells by clathrin-mediated endocytosis. Therapeutic approaches may target proteins involved in EX uptake. Note that there are numerous ways that EXs can be taken up by recipient cells including caveolin-mediated endocytosis, lipid raft-mediated endocytosis, micropinocytosis, phagocytosis, and membrane fusion [70]. (**D**). There are numerous ways to load therapeutic cargos into EXs and deliver them to target cells in the brain, as described in the text. The figure was created with BioRender.com https://app.biorender.com/ (accessed on 24 March 2023).

**Table 1 biomedicines-11-01187-t001:** Advantages and disadvantages, type of cargo that can be loaded, and whether a therapeutic cargo has been used to treat PD for each loading method.

Loading Method	Advantages	Disadvantages	Type of Cargo Loaded	Therapeutic Cargo for PD
Incubation	Simple Minimal effects on EX structure	Low loading capacity Hard to control the amount of cargo loaded	Small drugs, nucleic acids, proteins, peptides	BDNF, co-enzyme Q10, curcumin, DA
Transfection	Easy	Low loading efficiency Possible introduction of impurities	Nucleic acids, proteins, peptides	Catalase mRNA, Tom40
Sonication	Simple High loading capacity	Produces damage to EX membrane	Small drugs, proteins, peptides	None
Electroporation	High loading efficiency	Produces damage to EX membrane Potential to induce EX aggregation	Small drugs, nucleic acids, nanoparticles	shRNA-MC
Extrusion	High loading capacity	Produces damage to EX membrane	Small drugs, proteins, peptides	Catalase
Freeze-Thaw Method	Simple and effective	Potential to inactivate proteins Potential to induce EX aggregation Potential for liposome-EX fusion	Small drugs, proteins, peptides	None
Incubation with Membrane Permeabilizers	Easy High loading capacity	Hemolytic activity of saponin	Small drugs, proteins, peptides	None
Dialysis	Simple High loading capacity	Cellular uptake of dialysis-loaded EX is variable Potential to degrade proteins and peptides	Small drugs, nucleic acids, proteins, peptides	None

## Data Availability

Not applicable.

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
