# Peer review of "Potential for Therapeutic-Loaded Exosomes to Ameliorate the Pathogenic Effects of α-Synuclein in Parkinson’s Disease"

_biomedicines, 2023, doi:10.3390/biomedicines11041187_

Round 1

Reviewer 1 Report

This paper from Dr Rademacher is an interesting review describing into detail Exosomes (EXs) and their therapeutic applications for Parkinson's disease. The paper is well written and relevant for the scientific community.  I would suggest it for publication. 

Minor comment: the first sentence of the abstract appears disconnected from the second; I would suggest removing it and reformatting the abstract accordingly.

Author Response

Reviewer 1 Comments

Minor comment: the first sentence of the abstract appears disconnected from the second; I would suggest removing it and reformatting the abstract accordingly.

Author Response: I agree with the reviewer. The first sentence of the Abstract has been removed.

Reviewer 2 Report

Review of a manuscript “Potential for Therapeutic-Loaded Exosomes to Ameliorate the Pathogenic Effects of α-Synuclein in Parkinson’s Disease” by David J. Rademacher submitted to “Biomedicines”

Parkinson’s disease is the second prevalent after Alzheimer’s disease disorder for which there is no medications reversing the course of the illness. It puts a tremendous burden on patients, their family members, and the health care system. So further studies of its mechanisms are needed. The author reviewed the data on the role of extracellular vesicles in the pathogenesis of Parkinson’s disease, considering their role in intercellular communication and possibility of their use for developing next-generation delivery vehicles of therapeutic agents. This is an important field of biomedical research and the results discussed in the manuscript will be interesting for the readers of the “Biomedicines”.

The following corrections and additions should be done.

Abstract

“…which spreads α-syn pathology throughout the PD brain…” Recent studies show that GI and other organs are also involved in PD pathogenesis, so this point should be added.

2 Role of EVs in the Pathogenesis of Parkinson’s Disease

Lines 65-66. In vitro experiments provided the first evidence that newly synthesized monomeric and aggregated α-syn was released into the extracellular environment [30, 31]. The author should add here the citation to the following article: “Cell Responses to Extracellular α-Synuclein. Molecules. 2019 Jan 15;24(2):305. doi: 10.3390/molecules24020305.

3. Role of Microglia in the Pathogenesis of PD

Lines 93-103. “Microglia can enhance neuronal survival by releasing trophic factors, clearing debris, dead cells, and misfolded α-syn aggregates in PD [49-52]…. Activation of microglia induces an oxidative stress response, including the release of reactive oxygen species (ROS) and nitric oxide (NO)”

The author should find a consensus between positive and negative role of microglia. For example, by saying that microglia plays a controversial role and may be both beneficial and possess harmful effect.

 8. Strategies to Load EXs with Therapeutic Cargo

Lines 241-243.  “After EXs are isolated from tissues, body fluids, or cell culture medium by ultracentrifugation, gradient ultracentrifugation, co-precipitation, size exclusion chromatography, or field flow fractionation [115], they can be loaded with cargo.”

Isolated Exs may already contain some ingredients as cargo which may affect their use as carriers. The author should discuss what should be done to minimize this effect. 

Another point. When the author says “ultracentrifugation, gradient ultracentrifugation” he presumably means “differential or gradient ultracentrifugation”

 10. Conclusions.

Lines 507-508. “Pathogenic forms of α- syn are transferred via EXs to and from neurons, astrocytes, and microglia.” The author should be more specific explaining which pathogenic forms of α-syn he means.

 Figure 2. Some fonts on Figure 2 are too small and should be increased. For example. The words in the rectangular of Figure 2A are hardly seen.

 Legend to Figure 2.

Line 132 “Therapeutic approaches may target proteins involved in EX internalization/uptake. “The sense of this sentence is unclear. The author should be more specific, clearly explaining what he means.

Author Response

Reviewer 2 Comments

Abstract: “…which spreads α-syn pathology throughout the PD brain…” Recent studies show that GI and other organs are also involved in PD pathogenesis, so this point should be added.

Author response: I agree with the reviewer. There is strong evidence that α-syn propagates from the olfactory bulb and the gut then throughout the brain. The Abstract has been revised accordingly.

2 Role of EVs in the Pathogenesis of Parkinson’s Disease

Lines 65-66. In vitro experiments provided the first evidence that newly synthesized monomeric and aggregated α-syn was released into the extracellular environment [30, 31]. The author should add here the citation to the following article: “Cell Responses to Extracellular α-Synuclein. Molecules. 2019 Jan 15;24(2):305. doi: 10.3390/molecules24020305.

Author response: I agree. The citation has been added as per the reviewer’s request.

3 Role of Microglia in the Pathogenesis of PD

Lines 93-103. “Microglia can enhance neuronal survival by releasing trophic factors, clearing debris, dead cells, and misfolded α-syn aggregates in PD [49-52]…. Activation of microglia induces an oxidative stress response, including the release of reactive oxygen species (ROS) and nitric oxide (NO)” The author should find a consensus between positive and negative role of microglia. For example, by saying that microglia plays a controversial role and may be both beneficial and possess harmful effect.

Author response: I agree with the reviewer. It is accepted that microglia can have beneficial and harmful effects. I apologize for not making this point clearer. The text has been revised to emphasize this point.

8 Strategies to Load EXs with Therapeutic Cargo

Lines 241-243. Isolated Exs may already contain some ingredients as cargo which may affect their use as carriers. The author should discuss what should be done to minimize this effect. 

Author response: I thank the reviewer for raising this point. After EXs are isolated, they are purified to remove unwanted materials. I have revised the manuscript text accordingly to clarify this point. Please note that I did not mean to imply that isolated EXs were devoid of cargo in the lumen or embedded in their phospholipid bilayer. The presence of material in the EX lumen and the phospholipid bilayer confers advantages with regard to the use of EXs as carriers. Indeed, the presence of cargo in the EX lumen and phospholipid bilayer is at least partially responsible for their tropism, the ability of EXs to readily cross the BBB, and immune resistance.

When the author says “ultracentrifugation, gradient ultracentrifugation” he presumably means “differential or gradient ultracentrifugation”

Author response: Correct. The text has been edited accordingly.

10 Conclusions.

Lines 507-508. “Pathogenic forms of α- syn are transferred via EXs to and from neurons, astrocytes, and microglia.” The author should be more specific explaining which pathogenic forms of α-syn he means.

Author response: I thank the reviewer for this comment. The manuscript text has been revised to clarify that pathogenic, misfolded forms of α-syn are transferred via EXs to and from neurons, astrocytes, and microglia.

Figure 2. Some fonts on Figure 2 are too small and should be increased. For example. The words in the rectangular of Figure 2A are hardly seen.

Author response: Thank you for bringing this to my attention. Figure 2 has been modified to ensure that the reader can readily see all of the text. The font size has been increased.

Legend to Figure 2.

Line 132 “Therapeutic approaches may target proteins involved in EX internalization/uptake. “The sense of this sentence is unclear. The author should be more specific, clearly explaining what he means.

Author response: Thank you for raising this point. The term “internalization” has been removed from the manuscript as it is confusing to say “internalization/uptake.” The term “uptake” accurately conveys my intended meaning.
